# Protein engineering expands the effector recognition profile of a rice NLR immune receptor

Juan Carlos De la Concepcion[1†], Marina Franceschetti[1†], Dan MacLean[2], Ryohei Terauchi[3,4], Sophien Kamoun[2], Mark J Banfield[1]*

[1]Department of Biological Chemistry, John Innes Centre, Norwich, United Kingdom; [2]The Sainsbury Laboratory, University of East Anglia, Norwich, United Kingdom; [3]Division of Genomics and Breeding, Iwate Biotechnology Research Center, Iwate, Japan; [4]Laboratory of Crop Evolution, Graduate School of Agriculture, Kyoto University, Kyoto, Japan

**Abstract** Plant nucleotide binding, leucine-rich repeat (NLR) receptors detect pathogen effectors and initiate an immune response. Since their discovery, NLRs have been the focus of protein engineering to improve disease resistance. However, this approach has proven challenging, in part due to their narrow response specificity. Previously, we revealed the structural basis of pathogen recognition by the integrated heavy metal associated (HMA) domain of the rice NLR Pikp (Maqbool et al., 2015). Here, we used structure-guided engineering to expand the response profile of Pikp to variants of the rice blast pathogen effector AVR-Pik. A mutation located within an effector-binding interface of the integrated Pikp–HMA domain increased the binding affinity for AVR-Pik variants in vitro and in vivo. This translates to an expanded cell-death response to AVR-Pik variants previously unrecognized by Pikp in planta. The structures of the engineered Pikp–HMA in complex with AVR-Pik variants revealed the mechanism of expanded recognition. These results provide a proof-of-concept that protein engineering can improve the utility of plant NLR receptors where direct interaction between effectors and NLRs is established, particularly where this interaction occurs via integrated domains.
DOI: https://doi.org/10.7554/eLife.47713.001

*For correspondence:
Mark.banfield@jic.ac.uk

†These authors contributed equally to this work

**Competing interests:** The authors declare that no competing interests exist.

## Introduction

Protein engineering offers opportunities to develop new or improved molecular recognition capabilities that have applications in basic research, health and agricultural settings. Protein resurfacing, where the properties of solvent-exposed regions are changed (often by mutation), has been used extensively in diverse areas from antibody engineering for clinical use to the production of more stable, soluble proteins for biotechnology applications (*Chapman and McNaughton, 2016*).

Intracellular nucleotide binding, leucine-rich repeat (NLR) receptors are key components of plant innate immunity pathways. They recognize the presence or activity of virulence-associated, host-translocated pathogen effector proteins and initiate an immune response (*Kourelis and van der Hoorn, 2018*; *Cesari, 2018*). Because they confer resistance to disease, plant NLRs are widely used in crop breeding programs (*Dangl et al., 2013*). However, the recognition spectra of plant NLRs tend to be very specific, and pathogens may delete detected effectors from their genome or evolve novel effector variants that are not detected by the NLRs to re-establish disease (*Yoshida et al., 2016*).

The potential of engineering NLRs to overcome these limitations, or to detect new effector activities, is emerging (*Rodriguez-Moreno et al., 2017*). Gain-of-function random mutagenesis has

achieved some success in expanding the activation sensitivity or effector recognition profiles of NLRs (*Segretin et al., 2014*; *Giannakopoulou et al., 2015*; *Harris et al., 2013*). An alternative strategy, in which NLRs perceive protease effectors through their activity on engineered host proteins, can lead to expanded recognition profiles (*Carter et al., 2018*; *Helm et al., 2019*; *Kim et al., 2016*). When detailed knowledge of the direct binding interfaces between an effector and an NLR are known, there is potential for protein resurfacing to modify interactions and thereby impact immune signaling.

Plant NLRs are modular proteins that are defined by their nucleotide-binding (NB-ARC) and leucine-rich repeat (LRR) domains, but that also have either an N-terminal coiled-coil (CC) or a Toll/Interleukin-1/Resistance-protein (TIR) signaling domain (*Takken and Goverse, 2012*). Many NLRs also contain non-canonical integrated domains (*Kroj et al., 2016*; *Sarris et al., 2016*; *Bailey et al., 2018*). Integrated domains are thought to be derived from ancestral virulence-associated effector targets that directly bind pathogen effectors (or host proteins; *Fujisaki et al., 2017*) or are modified by them, thereby initiating an immune response (*Cesari et al., 2014*; *Le Roux et al., 2015*; *Maqbool et al., 2015*; *Sarris et al., 2015*; *De la Concepcion et al., 2018*). These domains present an exciting target for protein engineering approaches to improve NLR activities. NLRs containing integrated domains (often called the 'sensor') typically function in pairs, requiring a second genetically linked NLR (the 'helper') for immune signaling (*Białas et al., 2018*; *Eitas and Dangl, 2010*).

Two rice NLR pairs, Pik and Pia, contain an integrated Heavy Metal Associated (HMA) domain in their sensor NLR that directly binds effectors from the rice blast pathogen *Magnaporthe oryzae* (also known as *Pycularia oryzae*) (*Maqbool et al., 2015*; *Cesari et al., 2013*; *Ortiz et al., 2017*; *Guo et al., 2018*). The integrated HMA domain in the sensor NLR Pik-1 directly binds the effector AVR-Pik. Co-evolutionary dynamics has driven the emergence of polymorphic Pik-1 HMA domains and AVR-Pik effectors in natural populations, and these display differential disease-resistance phenotypes (*Costanzo and Jia, 2010*; *Kanzaki et al., 2012*; *Yoshida et al., 2009*). The Pikp NLR allele only responds to the effector variant AVR-PikD, but the Pikm allele responds to AVR-PikD, AVR-PikE, and AVR-PikA. These phenotypes can be recapitulated in the model plant *Nicotiana benthamiana* using a cell-death assay, and are underpinned by differences in effector–receptor binding interfaces that lead to different affinities in vitro (*Maqbool et al., 2015*; *De la Concepcion et al., 2018*).

We hypothesized that by combining naturally occurring favorable interactions observed across different interfaces, as defined in different Pik-HMA/AVR-Pik structures (*Maqbool et al., 2015*; *De la Concepcion et al., 2018*), we could generate a Pik NLR with improved recognition profiles. Here, we graft an interface from Pikm onto Pikp by mutating two residues in Pikp (Asn261Lys, Lys262Glu), forming Pikp$^{NK-KE}$. This single-site mutation strengthens the cell-death response in *N. benthamiana* to AVR-PikD, and gains a Pikm-like response to AVR-PikE and AVR-PikA. We show that this gain-of-function phenotype correlates with increased binding affinity of the effectors by the Pikp$^{NK-KE}$–HMA domain in vitro and in vivo, and demonstrate that this mutation results in a Pikm-like structure for Pikp$^{NK-KE}$ when in complex with AVR-Pik effectors. Finally, we confirm that the newly engineered interface is responsible for the expanded response of Pikp$^{NK-KE}$ by mutation of the effectors.

This study serves as a proof-of-concept for the use of protein resurfacing by targeted mutation to develop plant NLR immune receptors with new capabilities. In the future, such approaches have the potential to improve disease resistance in crops.

## Results

### Structure-informed engineering expands Pikp-mediated effector recognition in *N. benthamiana*

By comparing protein interfaces in the structures of Pikp-HMA and Pikm-HMA bound to different AVR-Pik effectors (*Maqbool et al., 2015*; *De la Concepcion et al., 2018*), we hypothesized that we could engineer expanded effector recognition capabilities by point mutation of Pikp. We constructed a series of mutations in the previously identified interface 2 and interface 3 regions of Pik-HMA–AVR-Pik structures (*De la Concepcion et al., 2018*), swapping residues found in Pikm into Pikp (*Figure 1A*, *Figure 1—figure supplement 1*). We then screened these mutations for expanded

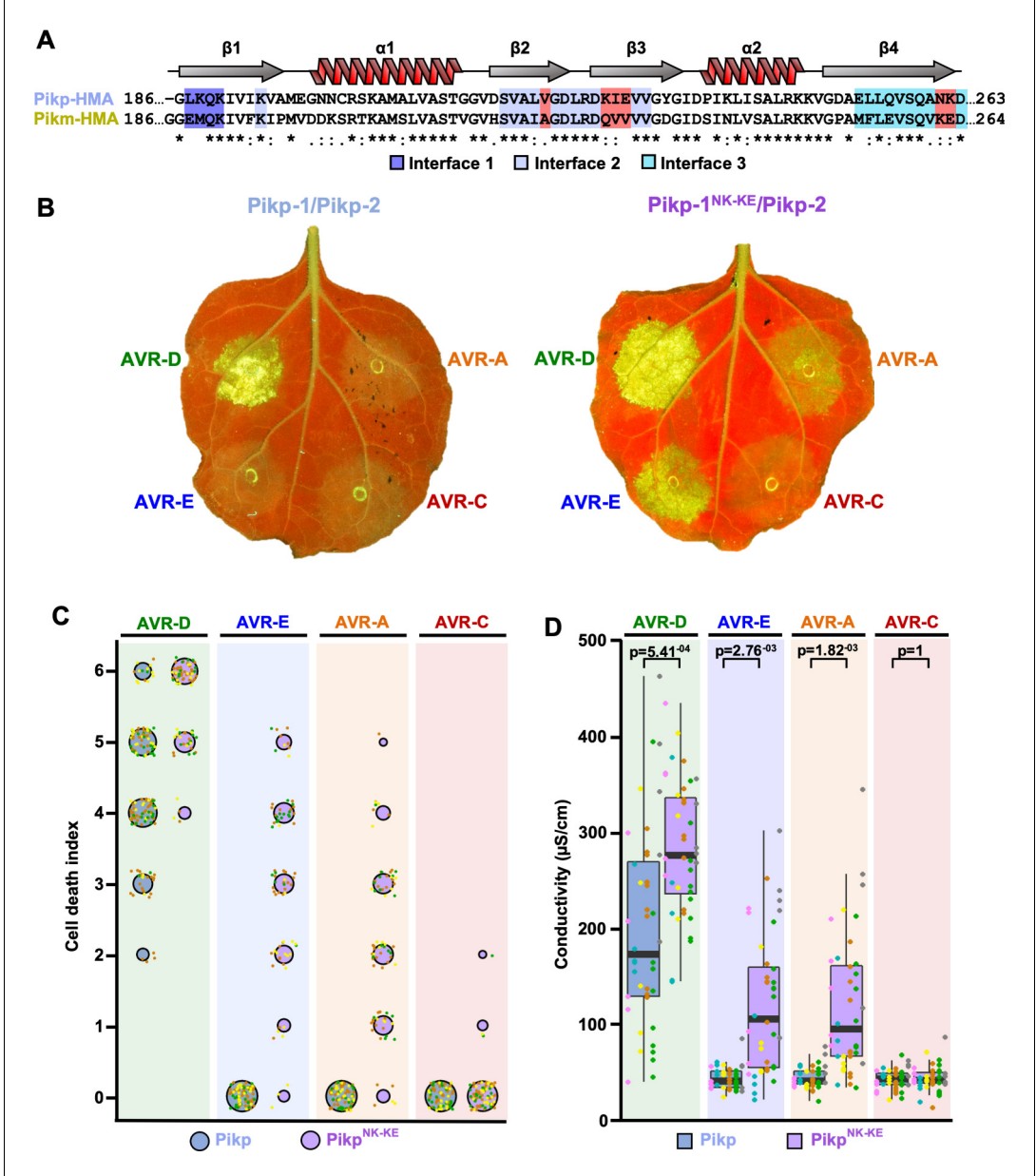

**Figure 1.** Structure-informed engineering expands Pikp-mediated effector recognition in *N. benthamiana*. (**A**) Sequence alignment of the Pikp-1 and Pikm-1 HMA domains. Secondary structure features of the HMA fold are shown above, and the residues that are located at binding interfaces are as colored. Key residues from interface 2 and interface 3 involved in this study are highlighted in red. (**B**) Representative leaf images showing Pikp- (left) or Pikp-1[NK-KE] (right)-mediated cell death in response to AVR-Pik variants as autofluorescence under UV light. (**C**) Autofluorescence intensity is scored as previously described (***Maqbool et al., 2015***; ***De la Concepcion et al., 2018***). Cell death assay scores are represented as dot plots for Pikp and Pikp[NK-KE] (blue and purple, respectively). For each sample, all of the data points are represented as dots with a distinct color for each of the three biological replicates; these dots are plotted around the cell death score for visualization purposes. The size of the centre dot at each cell death value is directly proportional to the number of replicates in the sample with that score. The total number of repeats was 80. Data for Pikp have been previously shown (***De la Concepcion et al., 2018***), but was acquired at the same time as those for Pikp[NK-KE]. The estimation methods used to visualize differences in the data sets are shown in ***Figure 1—figure supplement 3***. (**D**) Conductivity measurements showing ion leakage as a quantitative measure of cell death. The centre line represents the median, the box limits are the upper and lower quartiles, the whiskers extend to the largest value within Q1 – 1.5x the interquartile range (IQR) and the smallest value within Q3 + 1.5x IQR. All the data points are shown as dots with distinct colors for each biological replicate. For each experiment, six biological replicates with 5 or 10 internal repeats were performed (total data points = 40). 'p' is the p-value obtained from statistical analysis and Tukey's HSD (honestly significant difference) test.

DOI: https://doi.org/10.7554/eLife.47713.002

The following source data and figure supplements are available for figure 1:

*Figure 1 continued on next page*

*Figure 1 continued*

**Source data 1.** Cell death scoring data used in the preparation of *Figure 1C*.
DOI: https://doi.org/10.7554/eLife.47713.009
**Source data 2.** Conductivity measurements used in the preparation of *Figure 1D* and *Figure 1—figure supplement 2B*.
DOI: https://doi.org/10.7554/eLife.47713.010
**Figure supplement 1.** Mutations at interface 2 of the Pikp-1 HMA domain compromise the response to AVR-Pik effectors.
DOI: https://doi.org/10.7554/eLife.47713.003
**Figure supplement 1—source data 1.** Cell death scoring data used in the preparation of *Figure 1—figure supplement 1A*.
DOI: https://doi.org/10.7554/eLife.47713.004
**Figure supplement 2.** Response of Pikp$^{NK-KE}$ to AVR-Pik effectors compared to that of Pikm.
DOI: https://doi.org/10.7554/eLife.47713.005
**Figure supplement 2—source data 1.** Cell-death scoring data used in the preparation of *Figure 1—figure supplement 2A*.
DOI: https://doi.org/10.7554/eLife.47713.006
**Figure supplement 3.** Estimation graphics for cell death, Pikp vs Pikp$^{NK-KE}$.
DOI: https://doi.org/10.7554/eLife.47713.007
**Figure supplement 4.** Estimation graphics for cell death, Pikm vs Pikp$^{NK-KE}$.
DOI: https://doi.org/10.7554/eLife.47713.008

effector recognition by monitoring cell death in a well-established *N. benthamiana* assay (*Maqbool et al., 2015*; *De la Concepcion et al., 2018*). We found that one double mutation in two adjacent amino-acid residues contained within interface 3, Asn261Lys and Lys262Glu (henceforth Pikp$^{NK-KE}$), induced cell death in response to AVR-PikE and AVR-PikA (*Figure 1*, *Figure 1—figure supplement 1A,B*). Mutations that cause loss of response to AVR-PikD are most likely to be compromised in effector binding. Western blots confirmed that all of the proteins were expressed in plants (*Figure 1—figure supplement 1C*).

We subsequently focused on this double mutant, and independently repeated the cell-death assay to ensure its robustness (*Figure 1B,C*, *Figure 1—figure supplement 3*). When looking at the intensity of cell death mediated by Pikp$^{NK-KE}$, as for the Pikm allele (*De la Concepcion et al., 2018*), we observed an effector hierarchy of AVR-PikD > AVR-PikE > AVR-PikA . Pikp$^{NK-KE}$ shows a comparable, but elevated (not statistically significant in the case of AVR-PikE), response to effector variants when compared to that of Pikm (*Figure 1—figure supplements 2* and *4*). The cell-death response to AVR-PikD mediated by Pikp$^{NK-KE}$ is similar but elevated when compared to that mediated by Pikp (*Figure 1B,C*). Pikp$^{NK-KE}$ does not show a response to the stealthy AVR-PikC variant. To obtain a quantitative measure of cell death, we performed ion-leakage assays (*Figure 1D*, *Figure 1—figure supplement 2*, *Supplementary file 1*). The results of these assays correlate well with the cell-death index scores based on autofluorescence described above.

We conclude that the single Asn261Lys/Lys262Glu (Pikp$^{NK-KE}$) mutation at interface 3 in the Pikp NLR expands this protein's recognition profile to include the effector variants AVR-PikE and AVR-PikA, similar to that observed for Pikm.

## The engineered Pikp$^{NK-KE}$-HMA mutant shows increased binding to effector variants in vivo and in vitro

We used yeast-2-hybrid (Y2H) and surface plasmon resonance (SPR) to determine whether the expanded Pikp$^{NK-KE}$ cell-death response in *N. benthamiana* correlates with increased binding affinity of the Pikp$^{NK-KE}$-HMA domain for AVR-Pik effectors.

As AVR-PikE and AVR-PikA showed some interaction with Pikp-HMA using these approaches (*Maqbool et al., 2015*; *De la Concepcion et al., 2018*), we tested interactions with Pikp$^{NK-KE}$-HMA side-by-side with interactions with wild-type Pikp-HMA. Using Y2H, we observed increases in growth and in blue coloration (both indicative of protein–protein interaction) for Pikp-HMA$^{NK-KE}$ with effectors AVR-PikE and, particularly, AVR-PikA when compared with Pikp-HMA (*Figure 2A*). This was accentuated with more stringent conditions (imposed by increasing concentration of Aureobasidin A). Unexpectedly, we also observed limited yeast growth for Pikp-HMA$^{NK-KE}$ with AVR-PikC at the lower stringency (*Figure 2A*). The unrelated *M. oryzae* effector AVR-Pii was used as a negative control. Expression of all proteins was confirmed in yeast (*Figure 2—figure supplement 1*).

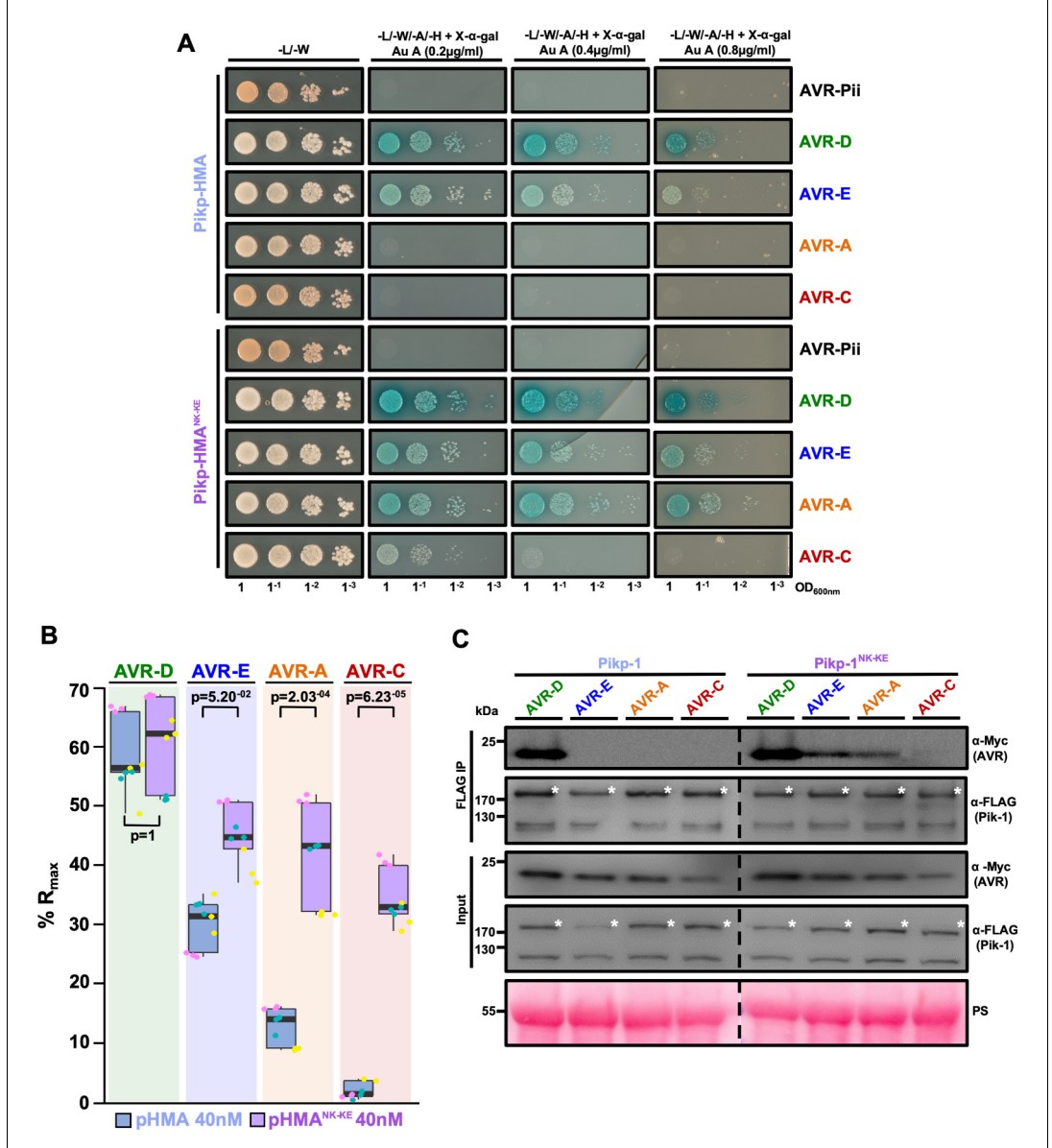

**Figure 2.** Pikp[NK-KE] shows increased binding to effector variants in vivo and in vitro when compared to wild type Pikp. (**A**) Yeast-two-hybrid assay of Pikp-HMA and Pikp-HMA[NK-KE] with AVR-Pik alleles. Control plates for yeast growth are on the left, with quadruple dropout media supplemented with X-α-gal and increasing concentrations of Aureobasidin A on the right for each combination of HMA/AVR-Pik. The unrelated *M. oryzae* effector AVR-Pii was used as a negative control. Growth and the development of blue coloration in the selection plate are both indicative of protein–protein interaction. HMA domains were fused to the GAL4 DNA binding domain, and AVR-Pik alleles to the GAL4 activator domain. Each experiment was repeated a minimum of three times, with similar results. (**B**) Box plots showing $\%R_{max}$, as measured by surface plasmon resonance, for Pikp-HMA and Pikp-HMA[NK-KE] with the AVR-Pik effectors alleles at an HMA concentration of 40 nM. Pikp-HMA and Pikp-HMA[NK-KE] are represented by blue and purple boxes, respectively. The centre line within each box represents the median, the box limits are the upper and lower quartiles, the whiskers extend to the largest value within $Q1 - 1.5 \times$ the interquartile range (IQR) and the smallest value within $Q3 + 1.5 \times$ IQR. All of the data points are represented as dots with distinct colors for each biological replicate. For each experiment, three biological replicates with three internal repeats were performed. 'p' is the p-value obtained from statistical analysis and Tukey's HSD. For results of experiments with 4 nM and 100 nM HMA protein concentration, see *Figure 2—figure supplement 2*. (**C**) Co-immunoprecipitation of full length Pikp-1 and Pikp-1[NK-KE] with AVR-Pik variants. N-terminally 4xMyc tagged AVR-Pik effectors were transiently co-expressed with Pikp-1:6xHis3xFLAG (left) or with Pikp-1[NK-KE]:6xHis3xFLAG (right) in *N. benthamiana*. Immunoprecipitates (IPs) obtained with anti-FLAG antiserum, and total protein extracts, were probed with appropriate antisera. The dashed line indicates a crop site on the same blot used to compose the figure. Each experiment was repeated at least three times, with similar results. The asterisks mark the Pik-1 band. PS = Ponceau Stain.

DOI: https://doi.org/10.7554/eLife.47713.011

*Figure 2 continued on next page*

*Figure 2 continued*

The following source data and figure supplements are available for figure 2:

**Source data 1.** Surface plasmon resonance measurements used in the preparation of *Figure 2B*.
DOI: https://doi.org/10.7554/eLife.47713.018
**Figure supplement 1.** Western blot confirming the accumulation of proteins in yeast.
DOI: https://doi.org/10.7554/eLife.47713.012
**Figure supplement 2.** Binding of the Pikp-HMA[NK-KE] domain to the AVR-Pik effectors is consistently greater than that of Pikp-HMA.
DOI: https://doi.org/10.7554/eLife.47713.013
**Figure supplement 2—source data 1.** Surface plasmon resonance measurements used in the preparation of *Figure 2—figure supplement 2*, left panel.
DOI: https://doi.org/10.7554/eLife.47713.014
**Figure supplement 2—source data 2.** Surface plasmon resonance measurements used in the preparation of *Figure 2—figure supplement 2*, right panel.
DOI: https://doi.org/10.7554/eLife.47713.015
**Figure supplement 3.** Binding of the Pikp-HMA[NK-KE] domain to the AVR-Pik effectors is consistently greater than that of the Pikm-HMA domain.
DOI: https://doi.org/10.7554/eLife.47713.016
**Figure supplement 3—source data 1.** Surface plasmon resonance measurements used in the preparation of *Figure 2—figure supplement 3*.
DOI: https://doi.org/10.7554/eLife.47713.017

Next, we produced the Pikp-HMA[NK-KE] domain protein via overexpression in *E. coli* and purified it to homogeneity using well-established procedures for these domains (see Materials and methods; *Maqbool et al., 2015*; *De la Concepcion et al., 2018*). Using SPR, we measured the binding affinity of the Pikp-HMA[NK-KE] domain (alongside both wild-type Pikp-HMA and Pikm-HMA) to AVR-Pik effectors (*Figure 2B*, *Figure 2—figure supplements 2* and *3*). Response units (RU) were measured following the injection of Pik-HMAs at three different concentrations, after capturing AVR-Pik effectors on a Biacore NTA chip. RUs were then normalized to the theoretical maximum response ($R_{max}$), assuming a 2:1 interaction model for Pikp-HMA and Pikp-HMA[NK-KE], and a 1:1 interaction model for Pikm-HMA, as previously described (*De la Concepcion et al., 2018*). These data showed an increased binding of Pikp-HMA[NK-KE] to AVR-PikE and AVR-PikA compared to that of wild-type Pikp (*Figure 2B*, *Figure 2—figure supplement 2*, *Supplementary file 2*). The binding of Pikp-HMA[NK-KE] to the AVR-Pik effectors was also greater than that of Pikm-HMA (*Figure 2—figure supplement 3*, *Supplementary file 2*), correlating with the results of cell-death assays (*Figure 1—figure supplement 2*). In both cases, statistical analysis has been carried out for the 40 nM analyte (Pik-HMA) data as a representative concentration. Although neither Pikp-HMA nor Pikm-HMA domains show binding to AVR-PikC by SPR, we observe a gain-of-binding of this effector variant with Pikp-HMA[NK-KE] (*Figure 2B*, *Figure 2—figure supplements 2* and *3*, *Supplementary file 2*).

These results show that the Pikp-HMA[NK-KE] mutant has a higher binding affinity for effectors AVR-PikE and AVR-PikA than does the wild-type protein. This suggests that the increased binding affinity to the HMA domain correlates with the expanded cell-death response in planta (*Figure 1B*).

## The engineered Pikp[NK-KE] mutant expands the association of full-length Pik-1 with effector variants in planta

In addition to interaction with the isolated HMA domain, we tested whether the Asn261Lys/Lys262-Glu mutant could expand effector variant binding in the context of the full-length NLR. After generating the mutant in the full-length protein, we co-expressed either Pikp-1 or Pikp-1[NK-KE] with the AVR-Pik effector variants in *N. benthamiana*, followed by immunoprecipitation and western blotting to determine effector association.

AVR-PikD shows a robust association with Pikp-1. However, although we observe limited binding to the isolated Pikp-HMA domain in Y2H and SPR screens, we did not detect association of AVR-PikE and AVR-PikA with the full-length Pikp-1 in planta (*Figure 2C*). By contrast, we observe clear association of AVR-PikE and AVR-PikA with the Pikp-1[NK-KE] mutant, albeit with reduced intensity compared to AVR-PikD, correlating with the hierarchical cell-death response observed in planta (*Figure 1C*).

We also detect a very low level of association between full-length Pikp-1[NK-KE] and AVR-PikC (*Figure 2C*). However, co-expression of Pikp-1[NK-KE] and AVR-PikC does not result in macroscopic cell death in *N. benthamiana* (*Figure 1C*).

These results show that effector variant association with full-length Pikp-1 and Pikp-1[NK-KE] correlates with the in planta cell-death response (*Figure 1C*).

## The effector-binding interface in the Pikp[NK-KE] mutant adopts a Pikm-like conformation

Having established that the Pikp[NK-KE] mutant displays an expanded effector recognition profile compared to wild-type Pikp, we sort to determine the structural basis of this activity. To this end, we determined the crystal structures of Pikp-HMA[NK-KE] bound to AVR-PikD, and to AVR-PikE. We obtained samples of Pikp-HMA[NK-KE]/AVR-PikD and Pikp-HMA[NK-KE]/AVR-PikE complexes by co-expression in *E. coli* (described in the Materials and methods and by *De la Concepcion et al., 2018*). Each complex was crystallized (see Materials and methods) and X-ray diffraction data were collected at the Diamond Light Source (Oxford, UK) to 1.6 Å and 1.85 Å resolution, respectively. The details of X-ray data collection, structure solution, and completion are given in the Materials and methods and in *Table 1*.

The overall architecture of these complexes is the same as that observed for all Pik-HMA/AVR-Pik effector structures. A key interaction at interface 3, one of the previously defined Pik-HMA–AVR-Pik interfaces (*De la Concepcion et al., 2018*), involves a lysine residue (Lys262 in Pikp and Pikm) that

**Table 1.** Data collection and refinement statistics

| | Pikp[NK-KE]–AVR-PikD | Pikp[NK-KE]–AVR-PikE |
|---|---|---|
| **Data collection statistics** | | |
| Wavelength (Å) | 0.9763 | 0.9763 |
| Space group | $P\,2_1\,2_1\,2_1$ | $P\,2_1\,2_1\,2_1$ |
| **Cell dimensions** | | |
| $a, b, c$ (Å) | 29.79, 65.33, 75.86 | 66.46, 80.70, 105.58 |
| Resolution (Å)[*] | 32.80–1.60 (1.63–1.60) | 29.50–1.85 (1.89–1.85) |
| $R_{merge}$ (%)[#] | 8.1 (97.1) | 5.2 (75.1) |
| $I/\sigma I$[#] | 16.1 (2.6) | 31.0 (4.1) |
| Completeness (%)[#] | 100 (100) | 99.8 (97.8) |
| Unique reflections[#] | 20,294 (978) | 49337 (2963) |
| Redundancy[#] | 12.8 (13.3) | 18.3 (17.8) |
| $CC^{(1/2)}$ (%)[#] | 99.9 (86.6) | 100 (95.2) |
| **Refinement and model statistics** | | |
| Resolution (Å) | 32.82–1.60 (1.64–1.60) | 29.52–1.85 (1.90–1.85) |
| $R_{work}/R_{free}$ (%)[ˋ] | 19.7/23.2 (25.5/27.3) | 18.6/23.0 (29.1/35.0) |
| No. atoms (Protein) | 1277 | 3604 |
| B-factors (Protein) | 25.6 | 39.7 |
| **R.m.s. deviations**[ˋ] | | |
| Bond lengths (Å) | 0.009 | 0.012 |
| Bond angles (˚) | 1.5 | 1.4 |
| **Ramachandran plot (%)**[**] | | |
| Favored | 98.1 | 97.3 |
| Outliers | 0 | 0.2 |
| MolProbity Score | 1.41 (93rd percentile) | 1.59 (91st percentile) |

[*]The highest resolution shell is shown in parenthesis.
[#]As calculated by Aimless, [ˋ]As calculated by Refmac5, [**]As calculated by MolProbity.
DOI: https://doi.org/10.7554/eLife.47713.019

forms intimate contacts within a pocket on the effector surface (*Figure 3*). In order to position this lysine in the effector pocket, Pikp has to loop-out regions adjacent to this residue, compromising the packing at the interface (*De la Concepcion et al., 2018*, *Figure 3A [left panel], B [left panel], C*

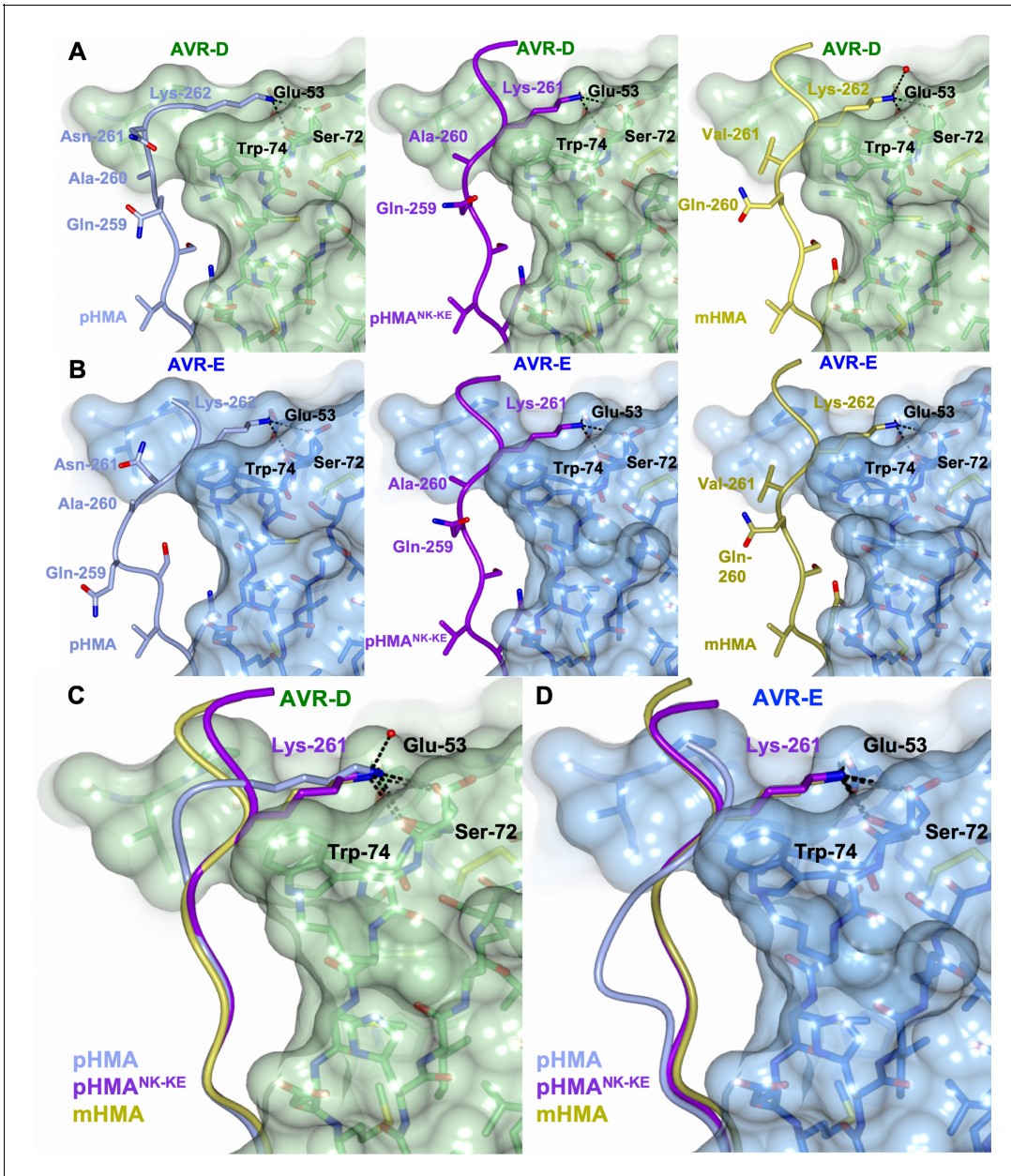

**Figure 3.** The Pikp[NK-KE]-HMA mutant adopts a Pikm-like conformation at the effector-binding interface. Schematic view of the different conformations adopted by Pikp-HMA, Pikp-HMA[NK-KE] and Pikm-HMA at interface 3 when in complex with AVR-PikD or AVR-PikE. In each panel, the effector is shown as sticks with the molecular surface also shown and colored as labeled. Pik-HMA residues are colored as labeled and shown in the Cα-worm with side-chain representation. (A) Schematic of Pikp-HMA (left), Pikp-HMA[NK-KE] (middle) and Pikm-HMA (right) bound to AVR-PikD. Important residues in the HMA–effector interaction are labeled as shown. (B) Schematic of HMA residues as for panel (A), but bound to AVR-PikE. (C) Superposition showing Pikp-HMA, Pikp-HMA[NK-KE] and Pikm-HMA chains (colored in blue, purple and yellow, respectively) bound to AVR-PikD. For clarity, only the Lys-261/262 side chain is shown. (D) Superposition as described before, but bound to AVR-PikE.

DOI: https://doi.org/10.7554/eLife.47713.020

The following figure supplement is available for figure 3:

**Figure supplement 1.** Interface 2 is essentially identical in the complexes comprising Pikp-HMA and Pikp-HMA[NK-KE] bound to AVR-PikD or AVR-PikE.

DOI: https://doi.org/10.7554/eLife.47713.021

and D). By contrast, in Pikm, where the position of the lysine is shifted one residue to the N-terminus, no looping-out is required to locate the lysine within the pocket (*Figure 3A [right panel], B [right panel], C and D*). In the Pikp$^{NK-KE}$ mutant, the position of this key lysine is shifted one residue to the N-terminus compared to that in the wild-type, so that it occupies the same position in the Pikp$^{NK-KE}$ sequence as in the Pikm sequence. In the crystal structures of Pikp-HMA$^{NK-KE}$ in complex with either AVR-PikD or AVR-PikE, we see that this region of the HMA adopts a Pikm-like conformation (*Figure 3A [middle panel], B [middle panel], C and D*), with no looping-out of the preceding structure. This confirms that in making the Pikp$^{NK-KE}$ mutant, we have resurfaced Pikp to have a more robust, Pikm-like interface in this region.

We found only limited structural perturbations at either of the other previously defined interfaces (interface 1 or 2 [*De la Concepcion et al., 2018*]) resulting from the binding of the AVR-PikD or AVR-PikE effectors to Pikp-HMA or Pikp-HMA$^{NK-KE}$ (*Figure 3—figure supplement 1*). We therefore conclude that the effects of the Pikp$^{NK-KE}$ mutant on protein function are mediated by altered interactions at interface 3.

## Mutation in AVR-Pik effectors at the engineered binding interface impacts the in planta response and in vivo binding

To further confirm that the engineered binding interface is responsible for the expanded recognition of AVR-PikE and AVR-PikA by Pikp$^{NK-KE}$, we used mutants in the effectors at interface 2 (AVR-PikD$^{H46E}$) and interface 3 (AVR-PikD,E,A$^{E53R}$), which have previously been shown to impact the interactions and in planta responses of wild-type NLR alleles (*De la Concepcion et al., 2018*). We tested whether these mutants affected the cell-death response in *N. benthamiana*, the interactions between effectors and Pikp-HMA$^{NK-KE}$ (using Y2H), and the interactions between effectors and full-length Pikp$^{NK-KE}$ (using in planta co-immunoprecipitation [co-IP]).

First, we investigated the impact of mutation at interface 2 using the AVR-PikD$^{H46E}$ mutant. We found that cell death in *N. benthamiana* is essentially blocked when co-expressing either Pikp or Pikp$^{NK-KE}$ with this mutant (AVR-PikD$^{H46E}$), suggesting that the engineered NLR is still reliant on this interface for response (*Figure 4A*, *Figure 4—figure supplement 1*). Intriguingly, Y2H shows that the AVR-PikD$^{H46E}$ mutant displays some interaction with Pikp-HMA$^{NK-KE}$ (*Figure 4B*), similar to this mutant's interaction with Pikm-HMA (*De la Concepcion et al., 2018*), although this interaction is barely observed by co-IP with the full-length NLR (*Figure 4C*).

Second, we investigated the impact of mutations at interface 3 using the Glu53Arg (E53R) mutant in AVR-PikD, AVR-PikE and AVR-PikA. We found that the AVR-PikD$^{E53R}$ mutant has essentially no effect on recognition of the effector by Pikp$^{NK-KE}$ in *N. benthamiana*, and little or no effect on the interaction with Pikp-HMA$^{NK-KE}$ or full-length Pikp$^{NK-KE}$ (*Figure 4A,B,C*, *Figure 4—figure supplement 1*). By contrast, the equivalent mutation in AVR-PikE and AVR-PikA restricted the cell-death response in *N. benthamiana*, reduced the binding to Pikp-HMA$^{NK-KE}$ in Y2H (as shown by the reduced blue coloration) and produced a less intense band for the effector following Pikp$^{NK-KE}$ co-IP (*Figure 4A,B,C*). The expression of all proteins in yeast was confirmed by western blot (*Figure 4—figure supplement 2*).

These results provide evidence to show that while interactions across interface 2 remain important for the Pikp$^{NK-KE}$ interaction with AVR-Pik effectors, it is the altered interaction at interface 3, as observed in the structures, that is responsible for the expanded recognition profile of this engineered mutant.

## Discussion

Plants, including food crops, are under continuous threat from pathogens and pests, and new solutions to control disease are required. Although largely elusive to date, engineered plant NLR-type intracellular immune receptors have the potential to improve disease-resistance breeding (*Dangl et al., 2013*; *Rodriguez-Moreno et al., 2017*). NLR-integrated domains are a particularly attractive target for protein engineering because they interact directly with pathogen effectors (or host effector targets). Further, where tested, their binding affinities in vitro correlate with in planta immunity phenotypes (*Maqbool et al., 2015*; *De la Concepcion et al., 2018*; *Guo et al., 2018*), allowing biochemical and structural techniques to inform NLR design directly.

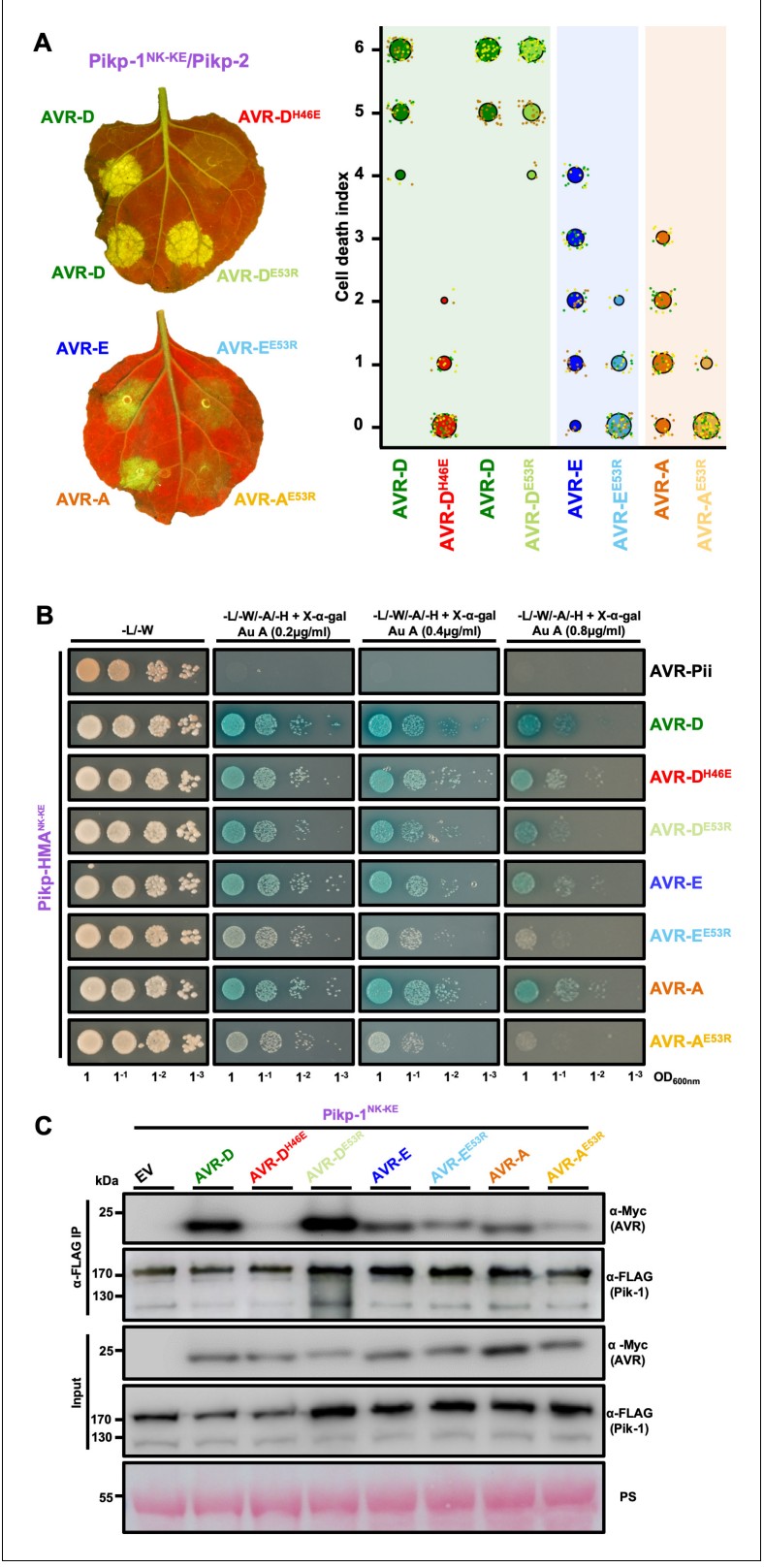

**Figure 4.** Mutation of AVR-Pik effectors at the engineered binding interface compromises binding and response. (**A**) (Left) A representative leaf image showing Pikp-1$^{NK-KE}$-mediated cell death induced by AVR-Pik variants and mutants as autofluorescence under UV light (the AVR-PikD$^{H46E}$ mutant is located at interface 2, whereas the AVR-PikD$^{E53R}$, AVR-PikE$^{E53R}$, and AVR-PikA$^{E53R}$ mutants are located at interface 3). Autofluorescence intensity is scored

*Figure 4 continued on next page*

*Figure 4 continued*

as in *Figure 1*. (Right) Pikp[NK-KE] cell-death assay quantification in the form of dot plots. For each sample, the data points are represented as dots with a distinct color for each of the three biological replicates; these dots are plotted around the cell death score for visualization purposes. The size of the central dot at each cell death value is proportional to the number of replicates of the sample with that score. The number of repeats was 90. (B) Yeast-two-hybrid assay of Pikp-HMA[NK-KE] with AVR-Pik variants and mutants. Control plate for yeast growth is on the left with quadruple dropout media supplemented with X-α-gal and increasing concentrations of Aureobasidin A on the right for each combination of HMA/AVR-Pik. The unrelated *M. oryzae* effector AVR-Pii was used as a negative control. Growth and the development of blue coloration in the selection plate are both indicative of protein–protein interaction. HMA domains were fused to the GAL4 DNA binding domain, and AVR-Pik alleles to the GAL4 activator domain. Each experiment was repeated a minimum of three times, with similar results. (C) Co-immunoprecipitation of full-length Pikp-1[NK-KE] with AVR-Pik variants and mutants. N-terminally 4xMyc tagged AVR-Pik effectors were transiently co-expressed with Pikp-1[NK-KE]:6xHis3xFLAG in *N. benthamiana* leaves. Immunoprecipitates (IPs) obtained with anti-FLAG antiserum, and total protein extracts, were probed with appropriate antisera. Each experiment was repeated at least three times, with similar results. The asterisks mark the Pik-1 band. PS = Ponceau Stain.

DOI: https://doi.org/10.7554/eLife.47713.022

The following source data and figure supplements are available for figure 4:

**Source data 1.** Cell-death scoring data used in the preparation of *Figure 4A*.
DOI: https://doi.org/10.7554/eLife.47713.025

**Figure supplement 1.** Estimation graphics for cell death induced by Pikp[NK-KE] with AVR-Pik variants and mutants.
DOI: https://doi.org/10.7554/eLife.47713.023

**Figure supplement 2.** Western blot analysis confirming the accumulation of proteins in yeast.
DOI: https://doi.org/10.7554/eLife.47713.024

Here, we show that the recognition profile of the rice NLR Pikp can be expanded to include different AVR-Pik variants by engineering the binding interface between these proteins. This strengthens the hypothesis that tighter binding affinity between effectors and integrated HMA domains correlates with increased immune signaling in plants. This was previously shown for both natural alleles of Pik (*Maqbool et al., 2015*; *De la Concepcion et al., 2018*) and for Pia (*Guo et al., 2018*), but is now also shown for an engineered NLR. We propose this may be a general model for integrated domains that directly bind effectors.

Natural variation in Pik NLRs has given rise to different effector recognition profiles, and contributions from different binding interfaces were suggested to underpin this phenotype (*De la Concepcion et al., 2018*). In particular, a more favorable interaction at one interface (interface 3) in Pikm, compared to that in Pikp, was concluded to have evolved to compensate for changes in binding at a different site (interface 2). Here, through mutation of residues in Pikp (forming Pikp[NK-KE]), we have combined favorable interfaces from Pikp and Pikm into a single protein. This has resulted in an expanded recognition phenotype for effector binding and response in planta. In vitro, the binding of Pikp[NK-KE]-HMA to effector variants is consistently higher than that of Pikm-HMA (*Figure 2—figure supplement 3*). This is likely to be underpinned by differences in a cluster of key residues within interface 2 (spatially equivalent residues Val222, Lys228, and Glu230 in Pikp, and Ala223, Gln229, and Val231 in Pikm) that form more energetically favorable contacts, including water-mediated hydrogen bonds. However, despite a positive trend (which is significant in some cases), the levels of cell death mediated by Pikm and Pikp[NK-KE] in planta are comparable (*Figure 1—figure supplements 2* and *4*).

Although the Pikp[NK-KE] mutant did not deliver a cell-death response in *N. benthamiana* to the stealthy AVR-PikC effector variant, it did show a gain-of-binding in vitro, as well as weak binding in Y2H and in in planta co-IP. It is surprising that the gain-of-binding for AVR-PikC in vitro, unlike that for AVR-PikA, does not correlate directly with gain of in vivo interactions or cell-death response in *N. benthamiana*. We hypothesize that this gain-of-binding for AVR-PikC observed in vitro is not sufficiently robust, especially in the context of the full-length NLR, for binding or for triggering immune signaling. Nevertheless, this work sets the scene for future interface engineering experiments that may further improve the response profiles of Pik NLRs to currently unrecognized effector variants. It also suggests that future work should test the disease resistance profile of *M. oryzae* strains carrying the different effector variants in rice expressing the engineered receptor.

The integrated HMA domain in the NLR RGA5 (the sensor of the Pia NLR pair in rice) binds to *M. oryzae* effectors AVR1-CO39 and AVR-Pia via a different interface, and it has been suggested that these binding sites are mutually exclusive (*Guo et al., 2018*). This raises the possibility that an HMA domain could be engineered to bind and respond to multiple effectors (*Guo et al., 2018*). Recently, the Pikp-HMA domain was shown to interact with AVR-Pia at the same interface as that used by the RGA5-HMA domain, and this probably underpins the partial resistance of rice plants to *M. oryzae* expressing AVR-Pia (*Varden et al., 2019*). This presents a starting point for the use of Pikp as a chassis for such studies. Although it remains to be seen whether any such resurfaced HMA domain can bind to multiple effectors, these studies suggest that this strategy has potential as a novel approach.

Plant breeding is required to provide new genetic solutions to disease resistance in crops. This is necessary to limit the environmental and social damage caused by pesticides, and to deal with changes in climate and in the globalization of agriculture that result in the spread of pathogens and pests into new environments (*Islam et al., 2016*; *Michelmore et al., 2017*; *Savary et al., 2019*). Classical breeding for disease resistance has been limited by issues such as linkage drag and hybrid incompatibility, as also seen in model plant species (*Bomblies et al., 2007*). Novel molecular approaches such as engineering 'decoys' (*Kim et al., 2016*) and protein resurfacing, as described here, combined with modern transformation (*Altpeter et al., 2016*) and breeding pipelines (*Watson et al., 2018*), offer the opportunity for more targeted approaches to breeding for disease resistance. These strategies will complement other emerging technologies in NLR identification (*Arora et al., 2019*) and NLR stacking (*Dangl et al., 2013*) as methods to develop improved crops for the future.

## Accession codes

Protein structures, and the data used to derive them, have been deposited at the Protein DataBank (PDB) with accession codes 6R8K (Pikp-HMA$^{NK-KE}$/AVR-PikD) and 6R8M (Pikp-HMA$^{NK-KE}$/AVR-PikE).

# Materials and methods

## Gene cloning

For in vitro studies, Pikp-HMA$^{NK-KE}$ (encompassing residues 186 to 263) was amplified from wild-type Pikp-HMA by introducing the mutations in the reverse primer, followed by cloning into pOPINM (*Berrow et al., 2007*). The wild-type Pikp-HMA, Pikm-HMA, and AVR-Pik expression constructs used in this study are as described in *De la Concepcion et al. (2018)*.

For Y2H, we cloned Pikp-HMA$^{NK-KE}$ (as above) into pGBKT7 using an In-Fusion cloning kit (Takara Bio USA), following the manufacturer's protocol. The wild-type Pikp-HMA domain in pGBKT7 and AVR-Pik effector variants in pGADT7 were generated as described in *De la Concepcion et al. (2018)*.

For protein expression in planta, the Pikp-HMA$^{NK-KE}$ domain was generated using site-directed mutagenesis by introducing the mutations in the reverse primer. This domain was then assembled into a full-length construct using Golden Gate cloning (*Engler et al., 2008*) and into the plasmid pICH47742 with a C-terminal 6xHis/3xFLAG tag. Expression was driven by the *Agrobacterium tumefaciens* Mas promoter and terminator. Full-length Pikp-1, Pikp-2, and AVR-Pik variants were generated as described in *De la Concepcion et al. (2018)*. All DNA constructs were verified by sequencing.

## Expression and purification of proteins for in vitro binding studies

pOPINM constructs encoding Pikp-HMA, Pikm-HMA and Pikp-HMA$^{NK-KE}$ were produced in *E. coli* SHuffle cells (*Lobstein et al., 2012*), using the protocol described in *De la Concepcion et al. (2018)*. Cell cultures were grown in autoinduction media (*Studier, 2005*) at 30˚C for 5–7 hr and then at 16˚C overnight. Cells were harvested by centrifugation and re-suspended in 50 mM Tris-HCl (pH 7.5), 500 mM NaCl, 50 mM glycine, 5% (vol/vol) glycerol, and 20 mM imidazole supplemented with EDTA-free protease inhibitor tablets (Roche). Cells were sonicated and, following centrifugation at 40,000xg for 30 min, the clarified lysate was applied to a Ni$^{2+}$-NTA column connected to an AKTA Xpress purification system (GE Healthcare). Proteins were step-eluted with elution buffer (50 mM Tris-HCl [pH7.5], 500 mM NaCl, 50 mM glycine, 5% (vol/vol) glycerol, and 500 mM imidazole)

and directly injected onto a Superdex 75 26/60 gel filtration column pre-equilibrated with 20 mM HEPES (pH 7.5) and 150 mM NaCl. Purification tags were removed by incubation with 3C protease (10 µg/mg fusion protein) followed by passing through tandem $Ni^{2+}$-NTA and MBP Trap HP columns (GE Healthcare). The flow-through was concentrated as appropriate and loaded onto a Superdex 75 26/60 gel filtration column for final purification and buffer exchange into 20 mM HEPES (pH 7.5) and 150 mM NaCl.

AVR-Pik effectors, with either a 3C protease-cleavable N-terminal SUMO or a MBP tag and with a non-cleavable C-terminal 6xHis tag, were produced in and purified from *E. coli* SHuffle cells as previously described (*Maqbool et al., 2015*; *De la Concepcion et al., 2018*). All protein concentrations were determined using a Direct Detect Infrared Spectrometer (Merck).

## Co-expression and purification of Pik-HMA and AVR-Pik effectors for crystallization

Pikp-HMA$^{NK-KE}$ was co-expressed with AVR-PikD or AVR-PikE effectors in *E. coli* SHuffle cells following co-transformation of pOPINM:Pikp-HMA$^{NK-KE}$ and pOPINA:AVR-PikD/E (which were prepared as described in *De la Concepcion et al., 2018*). Cells were grown in autoinduction media (supplemented with both carbenicillin and kanamycin), harvested, and processed as described in *De la Concepcion et al. (2018)*. Protein concentrations were measured using a Direct Detect Infrared Spectrometer (Merck).

## Protein–protein interaction: yeast-two-hybrid analyses

To detect protein–protein interactions between Pik-HMAs and AVR-Pik effectors in a yeast two-hybrid system, we used the Matchmaker Gold System (Takara Bio USA). Plasmid DNA encoding Pikp-HMA$^{NK-KE}$ in pGBKT7, generated in this study, was co-transformed into chemically competent Y2HGold cells (Takara Bio, USA) with the individual AVR-Pik variants or mutants in pGADT7 , as described previously (*De la Concepcion et al., 2018*). Single colonies grown on selection plates were inoculated in 5 ml of SD$^{-Leu-Trp}$ overnight at 30 °C. Saturated culture was then used to make serial dilutions of OD$_{600}$ 1, $1^{-1}$, $1^{-2}$, and $1^{-3}$, respectively. 5 µl of each dilution was then spotted on a SD$^{-Leu-Trp}$ plate as a growth control, and on a SD$^{-Leu-Trp-Ade-His}$ plate containing X-α-gal and supplemented with Aureobasidin A (Takara Bio, USA). Plates were imaged after incubation for 60–72 hr at 30 °C. Each experiment was repeated a minimum of three times, with similar results.

To confirm protein expression in yeast, total protein extracts from transformed colonies were produced by boiling the cells for 10 min in LDS Runblue sample buffer. Samples were centrifugated and the supernatant was subjected to SDS-PAGE gels prior to western blotting. The membranes were probed with anti-GAL4 DNA-BD (Sigma) for the HMA domains in pGBKT7 and with the anti-GAL4 activation domain (Sigma) antibodies for the AVR-Pik effectors in pGADT7.

## Protein–protein interaction: surface plasmon resonance

Surface plasmon resonance (SPR) experiments were performed on a Biacore T200 system (GE Healthcare) using an NTA sensor chip (GE Healthcare). The system was maintained at 25°C, and a flow rate of 30 µl/min was used. All proteins were prepared in SPR running buffer (20 mM HEPES [pH 7.5], 860 mM NaCl, 0.1% Tween 20). C-terminally 6xHis-tag AVR-Pik variants were immobilized on the chip, giving a response of 200 ± 100. The sensor chip was regenerated between each cycle with an injection of 30 µl of 350 mM EDTA.

For all the assays, the level of binding was expressed as a percentage of the theoretical maximum response (R$_{max}$) normalized for the amount of ligand immobilized on the chip. The cycling conditions were the same as those used in *De la Concepcion et al. (2018)*. For each measurement, in addition to subtracting the response in the reference cell, a further buffer-only subtraction was made to correct for bulk refractive index changes or machine effects (*Myszka, 1999*). SPR data were exported and plotted using R v3.4.3 (https://www.r-project.org/) and the function ggplot2 (*Wickham, 2009*). Each experiment was repeated a minimum of three times, including internal repeats, with similar results. The proteins used came from three independent preparations for the HMA domains and two independent preparations of the AVR-Pik effectors.

## Protein–protein interaction: in planta co-immunoprecipitation (co-IP)

Transient gene-expression in planta for Co-IP was performed by delivering T-DNA constructs within the *A. tumefaciens* GV3101 strain into 4-week-old *N. benthamiana* plants grown at 22–25°C with high-light intensity. *A. tumefaciens* strains carrying Pikp-1 or Pikp $-1^{NK-KE}$ were mixed with strains carrying the AVR-Pik effector, at $OD_{600}$ 0.2 each, in agroinfiltration medium (10 mM $MgCl_2$ and 10 mM 2-(N-morpholine)-ethanesulfonic acid (MES) [pH5.6]), supplemented with 150 μM acetosyringone. For detection of complexes in planta, leaf tissue was collected 3 days post infiltration (dpi), frozen, and ground to fine powder in liquid nitrogen using a pestle and mortar. Leaf powder was mixed with two times weight/volume ice-cold extraction buffer (10% glycerol, 25 mM Tris [pH 7.5], 1 mM EDTA, 150 mM NaCl, 2% w/v PVPP, 10 mM DTT, 1x protease inhibitor cocktail [Sigma], 0.1% Tween 20 [Sigma]), centrifuged at 4,200 g/4°C for 20–30 min, and the supernatant was passed through a 0.45 μm Minisart syringe filter. The presence of each protein in the input was determined by SDS-PAGE and western blot. Pik-1 and AVR-Pik effectors were detected by probing the membrane with anti-FLAG M2 antibody (SIGMA) and anti c-Myc monoclonal antibody (Santa Cruz), respectively. For immunoprecipitation, 1.5 ml of filtered plant extract was incubated with 30 μl of M2 anti-FLAG resin (Sigma) in a rotatory mixer at 4°C. After three hours, the resin was pelleted (800 g, 1 min) and the supernatant removed. The pellet was washed and resuspended in 1 ml of IP buffer (10% glycerol, 25 mM Tris [pH 7.5], 1 mM EDTA, 150 mM NaCl, 0.1% Tween 20 [Sigma]) and pelleted again by centrifugation as before. Washing steps were repeated five times. Finally, 30 μl of LDS Runblue sample buffer was added to the agarose and incubated for 10 min at 70°C. The resin was pelleted again, and the supernatant loaded onto SDS-PAGE gels prior to western blotting. Membranes were probed with anti-FLAG M2 (Sigma) and anti c-Myc (Santa Cruz) monoclonal antibodies.

## *N. benthamiana* cell-death assays

*A. tumefaciens* GV3101 vectors carrying Pikp-1, Pikm-1, or Pikp-1$^{NK-KE}$ were resuspended in induction media (10 mM MES [pH 5.6], 10 mM $MgCl_2$ and 150 μM acetosyringone) and mixed with Pikp-2 (or Pikm-2 in the Pikm cell-death assay), AVR-Pik effectors, and P19 at $OD_{600}$ 0.4, 0.4, 0.6 and 0.1, respectively. Four-week-old *N. benthamiana* leaves were infiltrated using a needleless syringe. Leaves were collected at 5 dpi to measure UV autofluorescence (a proxy for cell death) or ion leakage.

## Cell-death scoring: UV autofluorescence

Detached leaves were imaged at 5 dpi from the abaxial side of the leaves for UV fluorescence images. Photos were taken using a Nikon D4 camera with a 60 mm macro lens, ISO set 1600 and exposure ~10 secs at F14. The filter was a Kodak Wratten No.8 and white balance was set to 6250 degrees Kelvin. Blak-Ray longwave (365 nm) B-100AP spot light lamps were moved around the subject during the exposure to give an even illumination. Images shown are representative of three independent experiments, with internal repeats. The cell death index used for scoring was as presented previously (*Maqbool et al., 2015*). Dotplots were generated using R v3.4.3 (https://www.r-project.org/) and the graphic package ggplot2 (Wickham, H., 2009). The size of the centre dot at each cell death value is directly proportional to the number of replicates in the sample with that score. All individual data points are represented as dots.

## Cell-death scoring: ion leakage

For ion-leakage quantification, plants were infiltrated with the relevant constructs on two different leaves. At 5 dpi, leaves were detached and two leaf discs with a diameter of 8 mm (one disc per leaf spot) were collected and floated in 1.5 mL of Milli-Q water. Conductivity (μS/cm) was measured immediately after transferring the leaf disc to water (time zero) using a LAQUAtwin EC-33 conductivity meter (Horiba UK Ltd). Leaf discs were then incubated for 6 h at room temperature with gentle shaking before measuring the final conductivity. The assay was carried out in six biological replicates with a total of 40 technical replicates (2 discs x 40 plants). Conductivity data for each sample were exported and plotted using R v3.4.3 (https://www.r-project.org/) and the function ggplot2 (Wickham, H., 2009).

## Crystallization, data collection and structure solution

For crystallization, Pikp-HMA$^{NK-KE}$ in complex with AVR-PikD or AVR-PikE were concentrated following gel filtration. Sitting drop vapor diffusion crystallization trials were set up in 96 well plates, using an Oryx nano robot (Douglas Instruments, United Kingdom). Plates were incubated at 20°C, and crystals typically appeared after 24–48 hr. For data collection, all crystals were harvested from the Morpheus HT-96 screen (Molecular Dimensions), and snap-frozen in liquid nitrogen. Crystals used for data collection appeared from the following conditions: (i) Pikp-HMA$^{NK-KE}$/AVR-PikD (10 mg/ml), Morpheus HT-96 condition D4 (0.12 M alcohols [0.2 M 1,6-hexanediol, 0.2 M 1-butanol, 0.2 M 1,2-propanediol, 0.2 M 2-propanol, 0.2 M 1,4-butanediol, and 0.2 M 1,3-propanediol], 0.1 M Buffer system 1 [1 M imidazole, MES monohydrate (acid) (pH 6.5)], and 50% v/v precipitant mix 4 [25%v/v MPD, 25% v/v PEG 1000, 25% v/v PEG3350]); (ii) Pikp-HMA$^{NK-KE}$/AVR-PikE (15 mg/ml), Morpheus HT-96 condition A8 (0.06M divalents [0.3 M magnesium chloride hexahydrate, 0.3 M calcium chloride dihydrate]), 0.1M Buffer system 2 (sodium HEPES; MOPS [acid] [pH 7.5]), 37.5% v/v Precipitant mix 4 (25%v/v MPD, 25% v/v PEG 1000, 25% v/v PEG3350).

X-ray data sets were collected at the Diamond Light Source using beamline i03 (Oxford, UK). The data were processed using the xia2 pipeline (*Winter, 2010*) and CCP4 (*Winn et al., 2011*). The structures were solved by molecular replacement using PHASER (*McCoy et al., 2007*) and the coordinates of AVR-PikD and a monomer of Pikp-HMA from PDB entry 6G10. The final structures were obtained through iterative cycles of manual rebuilding and refinement using COOT (*Emsley et al., 2010*) and REFMAC5 (*Murshudov et al., 2011*), as implemented in CCP4 (*Winn et al., 2011*). Structures were validated using the tools provided in COOT and MOLPROBITY (*Chen et al., 2010*).

## Statistical analyses

Qualitative cell-death scoring from autofluorescence was analyzed using estimation methods (*Ho et al., 2019*) and visualized with estimation graphics using the besthr R library (*MacLean, 2019*). Briefly, in this process all autofluorescence (cell-death) scores in samples under comparison were ranked, irrespective of sample. The mean ranks of the control and test sample were taken and a bootstrap process was begun on ranked test data, in which samples of equal size to the experiment were replaced and the mean rank calculated. After 1000 bootstrap samples, rank means were calculated, a distribution of the mean ranks was drawn and its 2.5 and 97.5 quantiles calculated. If the mean of the control data is outside of these boundaries, the control and test means were considered to be different. Quantitative data from ion-leakage (cell-death) and SPR assays were analyzed by preparing a linear mixed effects model of sample on ion leakage/SPR and post-hoc comparisons performed for sample contrasts using Tukey's HSD method in the R package nlme (*Pinheiro and Bates, 2019*) and in lsmeans (*Lenth, 2016*).

## Acknowledgements

This work was supported by the BBSRC (grants BB/J004553, BB/P012574, BB/M02198X), the ERC (proposal 743165), the John Innes Foundation, the Gatsby Charitable Foundation, and JSPS KAKENHI 15H05779. We thank the Diamond Light Source, UK (beamline i03 under proposal MX13467) for access to X-ray data collection facilities. We also thank David Lawson and Clare Stevenson (JIC X-ray Crystallography/Biophysical Analysis Platform) for help with protein-structure determination and SPR, and Andrew Davis and Phil Robinson (JIC Bioimaging facilities) for photography.

## Additional information

### Funding

| Funder | Grant reference number | Author |
|---|---|---|
| Biotechnology and Biological Sciences Research Council | BB/J004553 | Kamoun S<br>Banfield MJ |
| Biotechnology and Biological Sciences Research Council | BB/P012574 | Kamoun S<br>Banfield MJ |

| Biotechnology and Biological Sciences Research Council | BB/M02198X | Franceschetti M<br>Kamoun S<br>Banfield MJ |
| --- | --- | --- |
| H2020 European Research Council | 743165 | Kamoun S<br>Banfield MJ |
| John Innes Foundation | | De la Concepcion JC<br>Franceschetti M<br>Banfield MJ |
| Gatsby Charitable Foundation | | Sophien Kamoun |
| Japan Society for the Promotion of Science | 15H05779 | Ryohei Terauchi |

The funders had no role in study design, data collection and interpretation, or the decision to submit the work for publication.

### Author contributions

Juan Carlos De la Concepcion, Formal analysis, Validation, Investigation, Methodology, Writing—original draft, Writing—review and editing; Marina Franceschetti, Formal analysis, Validation, Investigation, Methodology, Writing—review and editing; Dan MacLean, Conceptualization, Resources, Software, Funding acquisition, Validation, Visualization, Project administration, Writing—review and editing; Ryohei Terauchi, Conceptualization, Supervision, Funding acquisition, Project administration, Writing—review and editing; Sophien Kamoun, Conceptualization, Supervision, Funding acquisition, Validation, Writing—original draft, Project administration, Writing—review and editing; Mark J Banfield, Conceptualization, Resources, Software, Supervision, Funding acquisition, Validation, Visualization, Writing—original draft, Project administration, Writing—review and editing

### Author ORCIDs

Sophien Kamoun (ID) https://orcid.org/0000-0002-0290-0315
Mark J Banfield (ID) https://orcid.org/0000-0001-8921-3835

### Decision letter and Author response

Decision letter https://doi.org/10.7554/eLife.47713.035
Author response https://doi.org/10.7554/eLife.47713.036

# Additional files

### Supplementary files

• Supplementary file 1. Table of p-values for all pairwise comparisons of the ion-leakage data in *N. benthamiana*. Underlined values are those presented in the respective figures.
DOI: https://doi.org/10.7554/eLife.47713.026

• Supplementary file 2. Table of p-values for all pairwise comparisons of the SPR data including Pikp and Pikp$^{NK-KE}$. Underlined values are those presented in the respective figures.
DOI: https://doi.org/10.7554/eLife.47713.027

• Transparent reporting form
DOI: https://doi.org/10.7554/eLife.47713.028

### Data availability

Protein structures, and the data used to derive these, have been deposited at the Protein DataBank (PDB) with accession codes 6R8K (Pikp-HMANK-KE/AVR-PikD) and 6R8M (Pikp-HMANK-KE/AVR-PikE).

The following datasets were generated:

| Author(s) | Year | Dataset title | Dataset URL | Database and Identifier |
| --- | --- | --- | --- | --- |
| Juan Carlos De la Concepcion, Marina | 2019 | Complex of rice blast (Magnaporthe oryzae) effector | https://www.rcsb.org/structure/6R8K | Protein Data Bank, 6R8K |

| Franceschetti, Mark J Banfield | | protein AVR-PikD with an engineered HMA domain of Pikp-1 from rice (Oryza sativa) | | |
| Juan Carlos De la Concepcion, Marina Franceschetti, Mark J Banfield | 2019 | Complex of rice blast (Magnaporthe oryzae) effector protein AVR-PikE with an engineered HMA domain of Pikp-1 from rice (Oryza sativa) | https://www.rcsb.org/structure/6R8M | Protein Data Bank, 6R8M |

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
