## [Decision Letter]

Thank you for submitting your article "Protein engineering expands the effector recognition profile of a rice NLR immune receptor" for consideration by *eLife*. Your article has been reviewed by three peer reviewers, including Thorsten Nürnberger as the Reviewing Editor and Reviewer #1, and the evaluation has been overseen by Detlef Weigel as the Senior Editor.

The reviewers have discussed the reviews with one another and the Reviewing Editor has drafted this decision to help you prepare a revised submission.

Summary:

Engineering of plant NLR-type immune receptors for expanded effector recognition is an exciting and promising tool for future molecular breeding approaches. All reviewers agreed that this study provides a substantial step forward in our understanding on how to achieve this goal. While highly supportive of your work, the reviewers request some revisions, as listed below.

Essential revisions:

The quality of the Y2H data is not sufficient. Here, growth on triple or quadruple dropout medium supplemented with different 3AT concentrations that accentuate growth differences is required. Likewise, cell death phenotypes observed upon overexpression in *N. benthamiana* require proper quantification and statistical evaluation.

Reviewer #1:

This is a really nice and technically challenging complement to a number of papers of this laboratory. There is no doubt that it would make a nice contribution to *eLife*. My only concern is about quantification of cell death in this study. Here, the authors may resort to ion leakage measurements that are certainly better quantifiable than the cell death index chosen here.

Reviewer #2:

The manuscript at hand introduces an engineered variant of the rice NLR protein Pikp, the design of which was inspired by another rice NLR named Pikm. The authors argue that this Pikp variant, named Pikp^NK-KE^ binds the *Magnaporthe oryzae* effector variants AVR-PikA and AVR-PikE in addition to the cognate ligand AVR-PikD of wild-type Pikp. Binding of Pikp^NK-KE^ to AVR-PikA and AVR-PikE was shown in various ways including yeast-two-hybrid assay (Y2H), cell death assay in *N. benthamiana*, co-immunoprecipitation (Co-IP) and surface plasmon resonance (SPR) (the latter two using recombinant protein from *E. coli*, HMA domain only). In addition, the authors determined two crystal structures of the HMA domains of Pikp^NK-KE^ in complex with AVR-PikD and of Pikp^NK-KE^ in complex with AVR-PikE at 1.6 Å and 1.85 Å, respectively. The authors compare these crystal structures with previously published crystal structures of wild-type Pikp and Pikm bound to AVR-PikD and AVR-PikE. They conclude with using this structural information to develop effector variants that are not bound and unrecognized by the Pikp^NK-KE^ receptor, to which end they used cell death assay, Y2H and Co-IP.

As few studies exist that succeeded at engineering NLRs with novel specificity, the article is highly relevant to scientist within the field of plant immunity and beyond. I believe the study can be helpful and serve as a template to scientists who have similar endeavors of designing novel immune receptors. The novelty of the resolved structures lies in the mechanistic insight into effector binding of an engineered NLR.

For those reasons, I recommend publication of the manuscript in *eLife*.

However, major points need to be addressed before publication which are listed as follows:

1) Figure 2A and B: Why binding of AVR-PikC in case of Pikp^NK-KE^? Requires more substantial discussion with special focus on differences in yeast, tobacco and in vitro.

2) Figure 2—figure supplement 3: Discuss (by using crystal structure) why binding of pHMA^NK-KE^ is consistently higher than mHMA.

3) Figure 4B: Control missing. Why not show mutation of N46 in AVR-PikE and corresponding residue in AVR-PikA?

Reviewer #3:

This article describes structure-guided engineering of the Pikp NLR immune receptor from rice that recognizes the effector AVR-PikD from the blast fungus *Magnaporthe oryzae*. Engineering of the integrated HMA decoy domain of Pikm1 domain was performed to expand its recognition spectrum to the AVR-Pik allelic variants AVR-PikA and AVR-PikE. For this, the work builds on previous work on structural and functional characterization of the HMA domains of Pikm1 and PikP1.

The manuscript is very interesting, extremely well written and documented and provides a first example of structure guided engineering of an NLR immune receptor for extended recognition specificity. However, several weaknesses require revision:

- An important drawback of the study is that there is no analysis of the engineered Pikp-1^NK-KE^ mutant in rice plants to test whether it confers indeed an extended recognition specificity and resistance spectrum in transgenic rice.

- Another important weakness are the Yeast two hybrid data that are not convincing (cf specific comments).

- The *N. benthamiana* "cell death" assay should be improved by measuring fluorescence intensity on infiltrated spots and performing statistical analysis on these quantitative results.

- co-IP experiments with the isolated HMA domain (wt and mutant) would strengthen the study (c.f. specific comments).

Specific Comments:

Results and data, images and graphs "monitoring cell death in a well-established *N. benthamiana* assay": all images intended to show *N. benthamiana* cell death show autofluorescence, not cell death. Graphs are also based on autofluorescence induction and not cell death. Previous publications suggest that the cell death response is rather weak. Description of the experiments should be more accurate here and throughout the text. In addition, I do not understand why auto fluorescence is not quantified using image analysis software but interpreted visually (with a 6 level scoring system). Quantitative measurements combined with robust statistical analysis should be performed rather than scoring and qualitative interpretation of results from scoring. Differences between AVRPik-C and AVRPik-A on Pikp-1^NK-KE^ on the photo in Figure 1B seems quite subtle in particular if one assumes that this is among the most differential images that were obtained.

Figure 1—figure supplement 2 -Statistical test to conclude on difference is missing. Unclear what is meant by "qualitatively stronger response".

Figure 2A and 4B – Y2H experiments provide only weak support for the author's claims (in certain cases no support at all): growth differences between the different yeast clones are, in most cases, rather subtle and often don’t correlate with the activity and in vitro binding data. E.g. in Figure 2A, Pikp-HMA/ARVPikD and Pikp-HMA/AVR-PikA grow identically and Pikp-HMA/AVR-PikE even grow better. Pikp-HMA/AVR-PikE growth is stronger than Pikp-HMA NK/KE/AVR-PikE. In Figure 4B, all clone seem to grow the same. Differences in color are not always easy to appreciate and rely on subjective interpretation. I would like to see growth on triple or quadruple dropout medium supplemented with different 3AT concentrations that accentuate growth differences and allow to unambiguously differentiate between growth strength of different yeast clones. If authors prefer to rely on β-Galactosidase activity, they should provide quantitative data from enzyme assays. Also apparent inconsistencies between Y2H and SPR or *N. benthamiana* cell death assays should be addressed in the text. For Pikp-HMA^NK/KE^, a negative control (unrelated effector) should be provided since it seems to interact with all AVR-Pik constructs that were tested including some that are expected to not interact or to interact only very weakly (e.g. AVRPikD-H46E). It should be indicated on the figure that HMAs are fused to BD and Avr's to AD.

Figure 2C and Figure 4C: It would be interesting to see co-IP experiments with the isolated HMA domain (wt and mutant) to compare, on the one hand, to the other experiments with the isolated HMA domain (SPR and Y2H) and on the other to full-length Pikp1 variants.

Figure 2B and Figure 2—figure supplements 2 and 3 – SPR experiments: Binding of C is similar to binding of A and E. Statistical analysis should be performed to determine if there are differences. Why do authors not calculate affinities from SPR and discuss differences in affinities? One question to address would be, can the weak differences in affinity between C and A (or E) explain recognition specificity?

Figure 3—figure supplement 1: It looks to me as if the QtPISA interaction radar for Pikp-HMA^NK/KE^/AVR-PikE does not support that binding is biologically relevant. Area is highly similar to the one for Pikp-HMA/AVR-PikE. Please comment on this point.

Introduction, fourth paragraph: recognition does not necessarily rely on binding. Can be posttranslational modification (e.g. RRS1) or other mechanism.

Subsection “The engineered Pikp^NK-KE^ mutant expands association of full-length Pik-1 to effector variants in planta”: Replace "interact" by "associate with".

Subsection “The engineered Pikp^NK-KE^ mutant expands association of full-length Pik-1 to effector variants in planta”, last paragraph: Replace "binding" by "association".

Varden et al., 2019: This study is not peer-reviewed or properly published (Bioarchive paper) and therefore not be cited.

---

## [Author Response]

Essential revisions:The quality of the Y2H data is not sufficient. Here, growth on triple or quadruple dropout medium supplemented with different 3AT concentrations that accentuate growth differences is required. Likewise, cell death phenotypes observed upon overexpression in N. benthamiana require proper quantification and statistical evaluation.

Further details are provided below in the responses to each of the reviewers’ comments.

In brief, we have repeated our Y2H assays with increasing concentrations of Aureobasidin A, the antibiotic used in the Matchmaker Gold Y2H system (more stringent than 3AT in other systems). We screened different concentrations from 0.2µg/ml to 1µg/ml (0.2 µg/ml being the standard concentration in the Matchmaker protocol), our new figures show results for Aureobasidin A concentrations of 0.2, 0.4 and 0.8µg/ml. We hope you agree that this has improved the quality of the Y2H data (making it more robust), and that the results are clearer. Further, we have added a negative control, AVR-Pii, an unrelated effector from *M. oryzae*.

For the cell death assays, we do not think it appropriate to quantify the autofluorescence from the images as during acquisition the UV lamp is moved on a gantry and uniform light distribution is not guaranteed to allow this level of accuracy (we have added details to the Materials and methods about image acquisition). Therefore, we have chosen to develop the ion leakage assay to quantify cell death as an additional experiment. This has the added advantage of being another independent measure of immunity-related readouts in this study. As can be seen, the ion leakage assay compliments the scoring from autofluorescence very well. This includes a comparison between Pikp and Pikm that was the subject of our previous manuscript published in Nature Plants (direct comparison not included here, although is available).

Regarding statistics, as the data from the cell death assays is categorical, we use estimation methods to highlight differences. For the quantitative data (ion leakage assays and SPR binding data), we have used statistical analysis and Tukey’s HSD, and adjusted the text of the manuscript to take the results into account where necessary. In recognition of his contribution to this part of the manuscript, we have added an author, Dan MacLean, to the manuscript.

Reviewer #1:This is a really nice and technically challenging complement to a number of papers of this laboratory. There is no doubt that it would make a nice contribution to eLife. My only concern is about quantification of cell death in this study. here, the authors may resort to ion leakage measurements that are certainly better quantifiable than the cell death index chosen here.

In response to this reviewer, and others, we have now performed ion leakage experiments from *N. benthamiana* tissue. The results are presented in the new Figure 1D, Figure 1—figure supplement 2, and Supplementary file 1. The main text and Materials and methods have been updated appropriately. As the cell death scoring (by autofluorescence) and ion leakage assays correlate very well for the experiments with the AVR-Pik variants, we did not extend the ion leakage assays to include the effector mutants described in Figure 4.

Reviewer #2:[…] Major points need to be addressed before publication which are listed as follows:1) Figure 2A and B: Why binding of AVR-PikC in case of Pikp^NK-KE^? Requires more substantial discussion with special focus on differences in yeast, tobacco and in vitro.

We have edited the discussion concerning the binding of AVR-PikC to Pikp^NK-KE^ as measured in vitro (SPR), when there is only weak apparent binding by Y2H (accentuated in new results) and by co-IP in planta. We were surprised by the SPR result (especially in comparison to AVR-PikA). We think it is best interpreted as a difference between working in vitro with the purified Pik-HMA domain only vs. in vivo with the HMA domains (tagged with the GAL4-DNA binding domain, yeast) and with the full-length NLRs (in plants).

2) Figure 2—figure supplement 3: Discuss (by using crystal structure) why binding of pHMA^NK-KE^ is consistently higher than mHMA.

We have added a section to the Discussion on this point, and also put the in vitro binding data in the context of the in planta results.

3) Figure 4B: Control missing. Why not show mutation of N46 in AVR-PikE and corresponding residue in AVR-PikA?

An N46E mutation in AVR-PikE produces the same protein as AVR-PikD^H46E^, so this is already included in the experiment. While it is indeed the case that we have not included the N46E mutation in the AVR-PikA background, we feel this is an unnecessary control. Given our other data, it is highly likely that this mutation would have the same effect as in the AVR-PikD background.

Reviewer #3:[…] The manuscript is very interesting, extremely well written and documented and provides a first example of structure guided engineering of an NLR immune receptor for extended recognition specificity. However, several weaknesses require revision:- An important drawback of the study is that there is no analysis of the engineered Pikp-1^NK-KE^ mutant in rice plants to test whether it confers indeed an extended recognition specificity and resistance spectrum in transgenic rice.

We appreciate the comment, but we consider this work is beyond the scope of the manuscript.

- Another important weakness are the Yeast two hybrid data that are not convincing (cf specific comments).

Please see comments below, and the responses elsewhere to address the quality of the Y2H.

- The N. benthamiana "cell death" assay should be improved by measuring fluorescence intensity on infiltrated spots and performing statistical analysis on these quantitative results.

As mentioned above, we don’t think that the way we measure the autofluorescence lends itself to quantification per say. We have performed ion leakage assays to strengthen the conclusions and present those new results here, along with statistical analyses (including using estimation methods to analyse the scored autofluorescence data).

- co-IP experiments with the isolated HMA domain (wt and mutant) would strengthen the study (c.f. specific comments).

Please see comments below.

Specific Comments:Results and data, images and graphs "monitoring cell death in a well-established N. benthamiana assay": all images intended to show N. benthamiana cell death show autofluorescence, not cell death. Graphs are also based on autofluorescence induction and not cell death. Previous publications suggest that the cell death response is rather weak. Description of the experiments should be more accurate here and throughout the text. In addition, I do not understand why auto fluorescence is not quantified using image analysis software but interpreted visually (with a 6 level scoring system). Quantitative measurements combined with robust statistical analysis should be performed rather than scoring and qualitative interpretation of results from scoring. Differences between AVRPik-C and AVRPik-A on Pikp-1^NK-KE^ on the photo in Figure 1B seems quite subtle in particular if one assumes that this is among the most differential images that were obtained.

The experiments are now described in more detail in the main text. As mentioned previously, our method for imaging fluorescence is not standardised in terms of light distribution, meaning it is not strictly useful for quantitative scoring. Having said this, we now include using estimation methods for analysis of this data in the revised manuscript. More importantly, as mentioned above, we have also performed ion leakage assays to add a quantitative measure of the cell death to our results, including statistical analysis.

Regarding the image in Figure 1B. We have not selectively picked one of the most differential images to show, it is just one example of the many repeats we have conducted to ensure we are reporting a robust phenotype. We feel it does show a clear difference in response of AVR-PikA vs. AVR-PikC.

Figure 1—figure supplement 2 -Statistical test to conclude on difference is missing. Unclear what is meant by "qualitatively stronger response".

Having performed the estimation methods analysis, we have revised our wording describing this data.

Figure 2A and 4B – Y2H experiments provide only weak support for the author's claims (in certain cases no support at all): growth differences between the different yeast clones are, in most cases, rather subtle and do often not correlate with the activity and in vitro binding data. E.g. in Figure 2A, Pikp-HMA/ARVPikD and Pikp-HMA/AVR-PikA grow identically and Pikp-HMA/AVR-PikE even grow better. Pikp-HMA/AVR-PikE growth is stronger than Pikp-HMA NK/KE/AVR-PikE. In Figure 4B, all clone seem to grow the same. Differences in color are not always easy to appreciate and rely on subjective interpretation. I would like to see growth on triple or quadruple dropout medium supplemented with different 3AT concentrations that accentuate growth differences and allow to unambiguously differentiate between growth strength of different yeast clones. If authors prefer to rely on β-Galactosidase activity, they should provide quantitative data from enzyme assays. Also apparent inconsistencies between Y2H and SPR or N. benthamiana cell death assays should be addressed in the text. For Pikp-HMA NK/KE, a negative control (unrelated effector) should be provided since it seems to interact with all AVR-Pik constructs that were tested including some that are expected to not interact or to interact only very weakly (e.g. AVRPikD-H46E). It should be indicated on the figure that HMAs are fused to BD and Avr's to AD.

Thanks for the suggestions. We have repeated our Y2H assays (with some modifications), and made several changes in the new figures:

1) We repeated the assays adding increasing concentrations of Aureobasidin A (we chose this antibiotic over 3AT because is more stringent and used with the Matchmaker Gold system). We hope you agree that this has improved the quality of the Y2H data (making it more robust), and that the results are clearer.

2) As suggested, we have added an unrelated rice blast effector, AVR-Pii, as a negative control in the Y2H.

3) We had indicated the fusion (AD or BD) in supplementary Western blots regarding yeast protein expression in the original submission. We have now also added this information to the relevant main manuscript figure legends.

4) As also requested by reviewer 2, we have added extra discussion to the text concerning the binding of AVR-PikC to Pikp^NK-KE^ as measured in vitro (SPR), when there is only weak binding as measured by Y2H and very limited interaction by co-IP in planta.

Figure 2C and Figure 4C: It would be interesting to see co-IP experiments with the isolated HMA domain (wt and mutant) to compare, on the one hand, to the other experiments with the isolated HMA domain (SPR and Y2H) and on the other to full-length Pikp1 variants.

We thought about such experiments in earlier stages of the project. However, we decided to focus on the full-length NLR when expressing in planta, as this is more similar to the native state of the interaction.

Figure 2B and Figure 2—figure supplements 2 and 3 – SPR experiments: Binding of C is similar to binding of A and E. Statistical analysis should be performed to determine if there are differences. Why do authors not calculate affinities from SPR and discuss differences in affinities? One question to address would be, can the weak differences in affinity between C and A (or E) explain recognition specificity?

As mentioned above, we have added extra discussion to the text concerning the binding of AVR-PikC to Pikp^NK-KE^ as measured in vitro (SPR), when there is only weak binding as measured by Y2H and very limited interaction by co-IP in planta. Statistical analysis for the SPR data has now been included and changes to the text made where relevant.

Determining absolute, accurate, binding affinities in SPR requires additional concentrations of the analyte, and saturated binding to be achieved. As for our previously published work (Nature Plants), we were most interested in comparative binding analysis between effector variants and mutants for which the Rmax analysis is sufficient. Further, we struggle to work with high (saturating) concentrations of some HMA domains in SPR, including Pikp-HMA and Pikp-HMA^NK-KE,^as they can “stick” non-specifically to the chip surface and affect data quality at elevated concentrations. For these reasons we favour the Rmax analysis.

Figure 3—figure supplement 1: It looks to me as if the QtPISA interaction radar for Pikp-HMA^NK/KE^/AVR-PikE does not support that binding is biologically relevant. Area is highly similar to the one for Pikp-HMA/AVR-PikE. Please comment on this point.

Having reviewed the interaction radars presented, we agree with the reviewer that these are not useful in these cases and we have removed them. The interaction radars are a simplistic view of the interfaces, and here they prove to be confusing.

Introduction, fourth paragraph: recognition does not necessarily rely on binding. Can be posttranslational modification (e.g. RRS1) or other mechanism.

We have added “or be modified by them” which covers this valid point.

Subsection “The engineered Pikp^NK-KE^ mutant expands association of full-length Pik-1 to effector variants in planta”: Replace "interact" by "associate with".

Replaced “interaction” by “association”, also in the subsection heading “The engineered Pikp^NK-KE^ mutant expands association of full-length Pik-1 to effector variants in planta”.

Subsection “The engineered Pikp^NK-KE^ mutant expands association of full-length Pik-1 to effector variants in planta”, last paragraph: Replace "binding" by "association".

Replaced.

Varden et al., 2019: This study is not peer-reviewed or properly published (Bioarchive paper) and therefore not be cited.

Citation updated as the paper is now accepted at JBC.